# Investigation of Nitrogen and Vacancy Defects in Synthetic Diamond Plates by Positron Annihilation Spectroscopy

**DOI:** 10.3390/ma16010203

**Published:** 2022-12-26

**Authors:** Marat Eseev, Ivan Kuziv, Aleksey Kostin, Igor Meshkov, Aleksey Sidorin, Oleg Orlov

**Affiliations:** 1Northern (Arctic) Federal University, 17 Severnoi Dviny emb., 163002 Arkhangelsk, Russia; 2Joint Institute for Nuclear Research, 6 Joliot-Curie Str., 141980 Dubna, Russia

**Keywords:** NV-centers, positron annihilation spectroscopy, IR-spectroscopy, defects, vacancy, diamonds, unbreakable control

## Abstract

Currently, diamonds are widely used in science and technology. However, the properties of diamonds due to their defects are not fully understood. In addition to optical methods, positron annihilation spectroscopy (PAS) can be successfully used to study defects in diamonds. Positrons are capable of detecting vacancies, and small and large clusters of vacancies induced by irradiation, by providing information about their size, concentration, and chemical environment. By mapping in the infrared (IR) range, it is possible to consider the admixture composition of the main inclusions of the whole plate. This article presents the results of a study of defects in synthetic diamond plates, one of which was irradiated by electrons. It presents data about the distribution of the defect concentration obtained by Infrared spectroscopy. PAS with a monochromatic positron beam can be used as a non-destructive technique of detecting defects (vacancy) distribution over the depth of diamond plates.

## 1. Introduction

The diamond is the most promising wide band semiconductor [1] due to its cubic crystal structure and strong covalent bonds of carbon atoms, as well as its record high atomic density. However, many unique properties and prospects for high-tech applications of diamonds [2] are determined by the presence and concentration of various types of its lattice defects.

At the crystal lattice level, such defects can be zero-dimensional, one-dimensional, two-dimensional or three-dimensional. Zero-dimensional defects include point defects such as vacancies, impurity substitutions, and introductions. The presence of defects in diamond drastically affects its physical properties, the mechanical, thermophysical, electromagnetic and quantum properties of diamond change. 

In practical terms, the presence and concentration of defects can be both negative and positive factors. In the first case, their detection helps to cull products and improve the technological processes of diamond manufacturing. In the second case, varying the concentration of defects allows for obtaining unique diamond properties. One such unique property of diamonds is the possibility to manipulate spins of single atoms related to impurity defects and to read them with the help of optical methods [3].

The methods of manipulating spins in combination with methods of growing high-quality diamonds are already leading to the creation of quantum devices and applications such as resonant Forster energy transfer [4] and single-spin magnetometry [5], as well as new methods for obtaining images with unsurpassed resolution [6,7,8]. The interest in defects in diamonds and the mechanisms of their transformation seems to be relevant from the point of view of fundamental science and for the development of technologies for obtaining high-pressure/high-temperature (HPHT) and chemical vapor deposition (CVD) diamond single crystals with specified properties.

Defects of the vacancy type are the most interesting of all crystal lattice defects. One such defect is the nitrogen-vacancy-center (NV-center). The NV-center is a type of point defect in a diamond in which one carbon atom in the crystal lattice of a diamond is replaced by a nitrogen atom, and the neighboring lattice node remains vacant. Such an NV-center was first identified by Du Pre in 1965 [9], and the first important optical characteristics of such a center were described by Davis and Hamer in 1976 [10]. The first work in which single NV-centers were observed was published in 1997 [11]. After the discovery of single negatively charged centers, it was possible to demonstrate the photostable generation of single photons [12,13], which allowed the use of NV-centers in the implementation of quantum optical networks, as well as for electron spin readout [14,15], which defines such an NV-center as a possible solid state spin cubit, suitable for quantum information processing and applications in quantum sensing.

Known methods for studying natural and synthetic diamonds include the following:Vibrational spectroscopy (Raman and infrared spectroscopy, photoluminescence; absorption spectroscopy in the visible and UV regions).X-ray and neutron methods: small-angle scattering, topography, absorption spectroscopy (X-ray Absorption Fine Structure spectroscopy-XAFS).Nuclear physics methods: nuclear reaction method (Nuclear Reaction Analysis-NRA), and analysis of recoil nuclei (Elastic Recoil Detection Analysis-ERDA); positron spectroscopy.Secondary ion mass spectrometry (SIM) and thermal desorption.Microscopy and microprobe analysis.

Electron paramagnetic resonance (EPR) and optically detected magnetic resonance (ODMR) are used to diagnose NV centers.

We used a comparison of the generally accepted method of IR spectroscopy with PAS. The advantage of the latter is the high sensitivity of the method to single-crystal lattice defects at the level of 0.1 ppm and the possibility of scanning along the sample depth (z) (see, for example, [16]).

Today, positron annihilation spectroscopy, along with optical methods, is a relevant method for the study of NV-centers. This research method is extremely sensitive to the detection of defects at the crystal lattice level. The wide application of this method for the study of defects in semiconductors began in 1980 [17], and the first papers on the study of vacancy-type defects in diamonds were published in 2000 [18]. Positron annihilation spectroscopy can be realized by three methods: the method of positron annihilation lifetime spectroscopy method, the positron annihilation Doppler broadening spectroscopy method, and the angular correlation of radiation method [19,20].

The unique possibilities of non-destructive testing can be obtained using a monochromatic positron beam [21]. By changing its energy, it is possible to scan samples for the presence of defects in depth with extremely high accuracy. This method has been successfully used to analyze crystalline materials [22,23] for the appearance of vacancies under mechanical stress. PAS studies are used for semiconductors, metals, carbon nanostructures, and minerals (see, for example, [24,25]).

In this article, we studied the defects in nitrogen doped synthetic diamond plates which arise during electron irradiation. The main goal of the study focused on the determination of the type and concentration of defects. Infrared spectroscopy was used to determine the concentration of nitrogen and defects after irradiation, while positron annihilation spectroscopy was used to determine the presence of defects before and after irradiation. Subsurface defects in diamond plates and their depth profile were investigated using the Doppler annihilation line broadening method.

## 2. Experiment

The experiment was performed on two synthetic diamond plates (SAFU01 and SAFU02). The synthetic diamond plates were grown by HTHP technology using the temperature gradient method. The samples were then cut by laser with a side size of 4 × 4 mm and a height of 1.5 mm. The condition of the samples before electron irradiation was checked by infrared spectroscopy and the Doppler annihilation line broadening method. One sample (SAFU02) was left as a reference, the non-irradiated specimen. The SAFU01 sample was then irradiated with an energy of 10 MeV electrons up to a dose of 1.1 × 10^16^ cm^−2^. No defects after such a dose of irradiation were noted. Furthermore, the SAFU01 sample was again irradiated by electrons (3 MeV, 1 × 10^18^/cm^2^).

## 3. Methods

The samples were investigated by an infrared spectrometer, a SHIMADZU UV 3600i Plus (SHIMADZU, Kyoto, Japan), and by the Doppler annihilation line broadening technique of Positron Annihilation Spectroscopy. The measurement of the Doppler annihilation line broadening was performed using the ORTEC HPGe detector (model GEM25P4-70). Doppler spectroscopy was implemented on a positron beam facility with energy variation from 0.1 to 30 keV. The positrons were emitted directly from the source of the isotope ^22^Na. The flux intensity was 10^6^ e^+^/s, and the beam spot diameter was approximately 5 mm. The value of 2.5 × 10^5^ counts at the 511 keV annihilation peak.

The Doppler broadening of annihilation line method is used to detect concentrations of defects such as vacancies and vacancy accumulations. A signal from the annihilation of a trapped positron gives a broadening of the 511 keV line which is accordingly smaller than the one that would occur in case of annihilation with nucleus electrons. In other words, more defective sample gives a larger broadening of the 511 keV line. Each obtained spectrum was analyzed to calculate parameters S and W (Figure 1) using the SP-16K program [26]. Parameter S reflects annihilations of low-momentum electrons occurring in defects. It is sensitive to open volume defects such as vacancies and vacancy accumulations. Parameter S is defined as the ratio of the area below the central part (A_S_ within the energy range of 511 ± 1.03 keV) of the annihilation line to the total area in the range of this line (A within the energy range of 511 ± 10.3 keV). A larger value of parameter S means a larger concentration of defects in the sample. Parameter W reflects annihilations of high-momentum electrons and it provides information about the chemical environment of the defect. This parameter can indicate a change in the appearance or size of lattice defects. Parameter W is given as the ratio of the area under the wing part (A_W_—within the energy range from 511 ± 3.44 keV to 511 ± 10.3 keV) of the 511 keV line to the total area (A) marked by that line. 

The Doppler broadening of the annihilation line method is based on a slow monoenergetic positron beam with energies up to 30 keV located at the Joint Institute for Nuclear Research in Dubna, Russia.

## 4. Distribution Maps of IR Spectroscopy

In this paragraph, a yellow diamond plate SAFU01 was studied. The distribution maps of the main defects were obtained using a MIKRAN-3 spectrometer with a photodetector cooled in liquid nitrogen in the IR range of wave numbers from 1000 to 1500 cm^−1^.

Before starting the experiment, the sample surface was cleaned with ethyl alcohol. The best area for mapping was selected. The size of the area was 2311 by 2889 µm. The scanning step was 41.275 µm. The beam size was corrected by the aperture, and the beam area was 50 by 50 μm.

The following are maps of the defect distribution. The vertical scale is transmittance registered as a percentage.

Figure 2 shows the IR spectrum of a yellow diamond with the distribution of C-defect (defect C, or donor nitrogen, is a single nitrogen atom that isomorphically replaces the carbon atom in the diamond lattice):

The concentration of C defects in this diamond is calculated by the formula [27]:(1)$$N_{C}\;\left(\mathrm{ppm}\right)=\left(25\;\pm \;2\right)\;\;{µ}_{1130}$$

The concentration of the C defect in diamond SAFU01 ranges from 59 to 189 ppm.

The concentration of the C defect in diamond SAFU02 ranges from 164 to 200 ppm.

Figure 3 shows the IR spectrum of yellow diamond with A-defect distribution (A the defect contains two paired nitrogen):

The concentration of nitrogen atoms in the form of A-defects in this diamond is calculated by the formula [28]:(2)$$N_{A}\;\left(\mathrm{ppm}\right)=\left(16.2\;\pm 1\right)\;\;{µ}_{1282}$$

The concentration of nitrogen atoms in the form of A-defects in diamond SAFU01 is 36 to 51 ppm.

The concentration of nitrogen atoms in the form of A-defects in diamond SAFU02 is 50 to 70 ppm.

Figure 4 shows the IR spectrum of yellow diamond with N^+^-defect distribution:

The peak at the Raman frequency of 1332 cm^−1^ means that some of the C defects have lost their fifth valence electron and ended up with a positively charged nitrogen ion, *N^+^*. This occurs during irradiation [29] (the $N^{+}$ defect is a C defect with a positively charged nitrogen ion). The concentration of C^+^ defects in this state is calculated by the formula [30]:(3)$$N^{+}\;\left(\mathrm{ppm}\right)=\left(5.5\;\pm \;1\right)\;\;{µ}_{1332}$$

The concentration of C^+^ defects in diamond SAFU01 is from 8 to 12 ppm.

The concentration of C^+^ defects in diamond SAFU02 is from 13 to 20 ppm.

## 5. Results and Discussion

Pictures of reference sample SAFU02, sample SAFU01 after electron irradiation (10 MeV, 1 × 10^16^/cm^2^) and sample SAFU01 after electron re-irradiation (3 MeV, 1 × 10^18^/cm^2^) are shown in Figure 5. The sample SAFU01 changed colors after electron irradiation (3 MeV, 1 × 10^18^/cm^2^).

The results of IR spectroscopy after and before irradiation are shown in Figure 6. The samples SAFU01 and SAFU02 were grown under the same conditions, but they had some differences in nitrogen concentration (SAFU01—220 ppm, SAFU02—198 ppm). After electron irradiation (3 MeV, 1 × 10^18^/cm^2^), the nitrogen concentration in the sample SAFU01 decreased from 220 ppm to 200 ppm. The concentration of neutral vacancies was 5.2 ppm, and the concentration of negative vacancies was 20 ppm.

For Positron Annihilation Spectroscopy, diamond plates were placed at the outlet of the positron beam from the accelerator for Doppler spectroscopy. The samples were irradiated with positrons having energies from 0.1 keV to 26 keV. The maximum penetration depth of the positron beam into the diamond plate was 2.4 μm. The total sum of counts for each energy was 250,000. The results of Doppler Broadening after first electron irradiation (10 MeV, 1 × 10^16^/cm^2^) are shown in Figure 7. The results of Doppler Broadening after second electron irradiation (3 MeV, 1 × 10^18^/cm^2^) are shown in Figure 8.

The result of the Doppler spectroscopy was the determination of the dependence of parameter S in the sample on the positron energy. Figure 7 shows that the reference sample SAFU02 and the sample SAFU01 after electron irradiation (10 MeV, 1 × 10^16^/cm^2^) do not have a significant difference in defects. However, they are not identical. The decrease of the S-parameter for energies before 10 keV is caused by thermalized positrons, which diffuse back to the surface where they annihilate with a higher S-parameter than those in the bulk. Above positron implantation energies of 12 keV, the S-parameter takes a constant value. This means that almost all positrons annihilate in the bulk without diffusing back to the surface. The SW plot in Figure 7 does not show the presence of new types of defects. An analysis of the Doppler plots of the annihilation line broadening in Figure 8 showed that the appearance of defects in the sample SAFU01 occurred after electron re-irradiation (3 MeV, 1 × 10^18^/cm^2^). In the near-surface layer, the differences between the sample SAFU01 before and after irradiation (3 MeV, 1 × 10^18^/cm^2^) start at a positron energy of 2 keV. This roughly corresponds to 20 nm. The S parameter then decreases with increasing positron energy. Above positron implantation energies of 6 keV, the S-parameter takes a constant value. There are significant differences between the samples starting from a depth of about 200 nm. The SW plot in Figure 8 shows the appearance of a new type of defect in the sample SAFU01 after irradiation (3 MeV, 1 × 10^18^/cm^2^). From a comparison of Figure 7 and Figure 8, it can be observed that the area where almost all positrons annihilate in the bulk without diffusing back to the surface shift from 380 nm to 170 nm. Irradiation by electrons (3 MeV, 1 × 10^18^/cm^2^) created a significant increase in defects in a synthetic diamond plate at depths of up to 2.4 µm in contrast to irradiation by electrons with other parameters (10 MeV, 1 × 10^16^/cm^2^), which did not create defects in a sample at the same depths.

In addition, the positron diffusion length was determined using the VEPFIT program [31]. A one-layer model was used for determining the positron diffusion length. The positron diffusion length of the SAFU01 sample before irradiation was 83.3 nm. After electron irradiation (3 MeV, 1 × 10^18^/cm^2^), the positron diffusion length of the SAFU01 sample was 67.9 nm. The positron diffusion length decreases after electron irradiation (3 MeV, 1 × 10^18^/cm^2^), which represents an increased number of defects below the surface.

## 6. Conclusions

In summary, the investigation has shown that the irradiation of a synthetic diamond plate by electrons (10 MeV, 1 × 10^16^/cm^2^) is not enough for formation vacancy, and at NV-centers at depths of up to 2.4 µm, such electron irradiation practically does not influence vacancy defects formation at the same depths. The electron irradiation (3 MeV, 1 × 10^18^/cm^2^) causes the formation of vacancies in the sample, which is shown by both infrared spectroscopy and positron annihilation spectroscopy, with significant vacancy formation concentrated at depths from 200 nm to 2400 nm. IR spectroscopy has shown that, after irradiation, the nitrogen concentration in the sample decreased (from 220 ppm to 200 ppm) and defects have appeared (neutral vacancies—5.2 ppm and negative vacancies—20 ppm). Distribution maps of IR spectroscopy showed the nature of the nitrogen defect distribution. After irradiation, the distribution map of single-nitrogen defect distribution became blurred. It is shown that positron annihilation spectroscopy is a sensitive method of detecting defects in samples of diamond plates and can be used alongside other optical research methods, such as infrared spectroscopy, as a method of detecting defect distribution over the depth of diamond plates.

## Figures and Tables

**Figure 1 materials-16-00203-f001:**
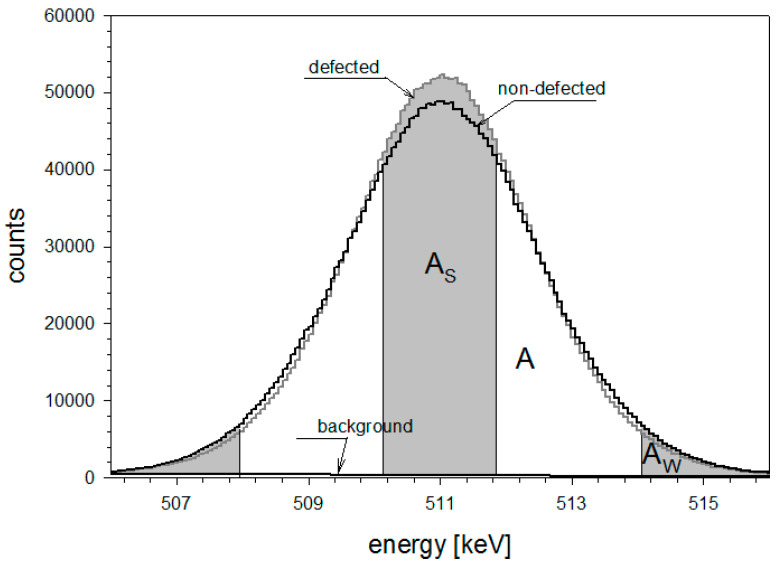
511 keV lines for defective and non-defective samples.

**Figure 2 materials-16-00203-f002:**
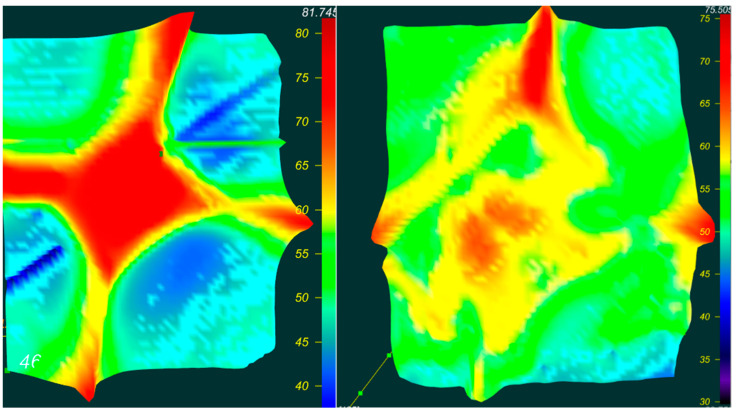
IR spectrum of C-defect distribution. ((**left**)—SAFU01, (**right**)—SAFU02).

**Figure 3 materials-16-00203-f003:**
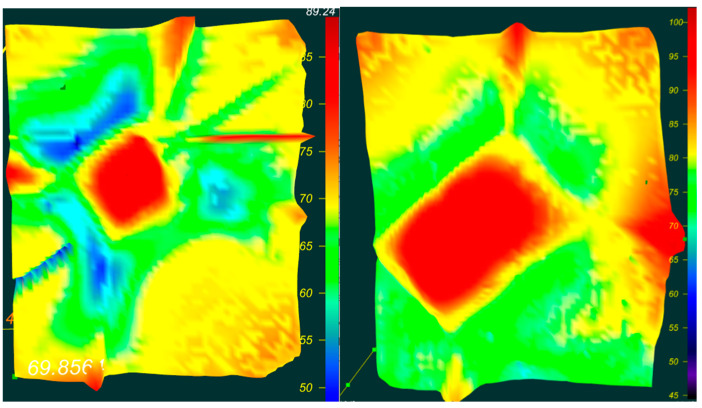
IR spectrum of A-defect distribution. ((**left**)—SAFU01, (**right**)—SAFU02).

**Figure 4 materials-16-00203-f004:**
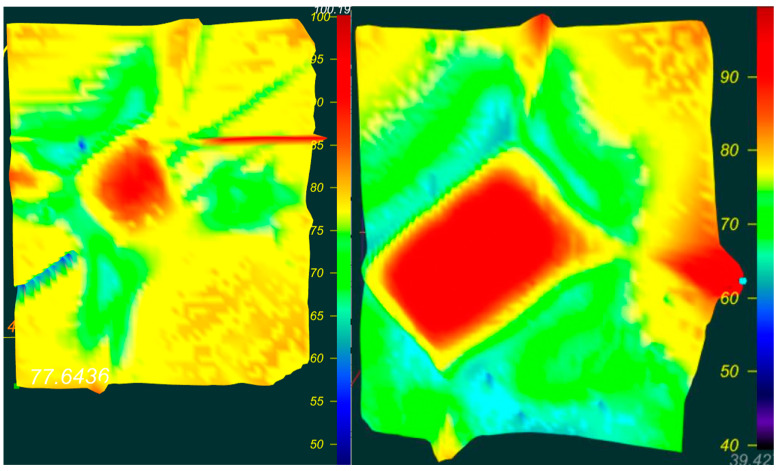
IR spectrum of N^+^-defect distribution. ((**left**)—SAFU01, (**right**)—SAFU02).

**Figure 5 materials-16-00203-f005:**
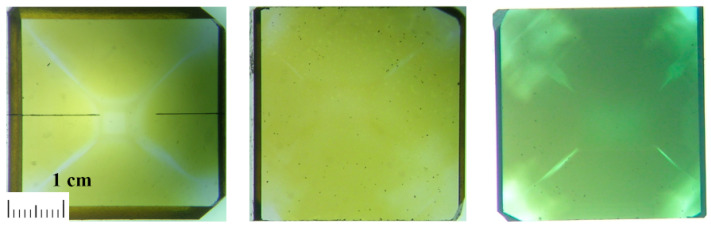
Picture of the samples ((**left**)—reference sample SAFU02, (**middle**)—sample SAFU01 after electron irradiation (10 MeV, 1 × 10^16^/cm^2^), (**right**)—sample SAFU01 after electron irradiation (3 MeV, 1 × 10^18^/cm^2^)).

**Figure 6 materials-16-00203-f006:**
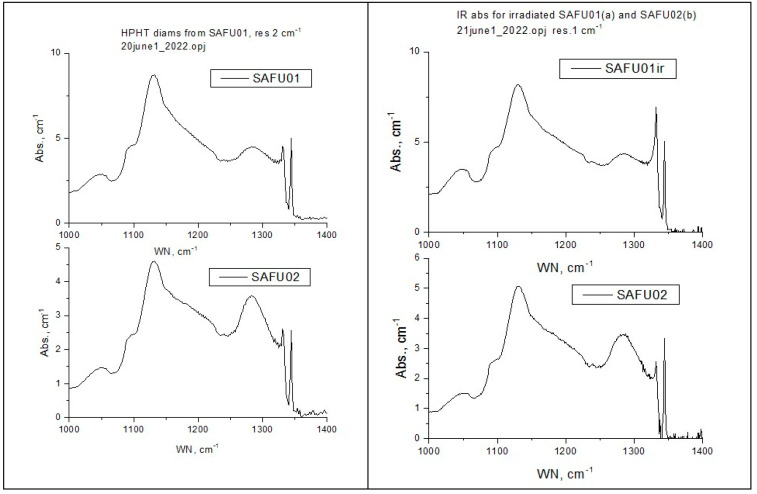
Results of IR-Spectroscopy ((**left column**)—before irradiation, (**right column**)—after irradiation).

**Figure 7 materials-16-00203-f007:**
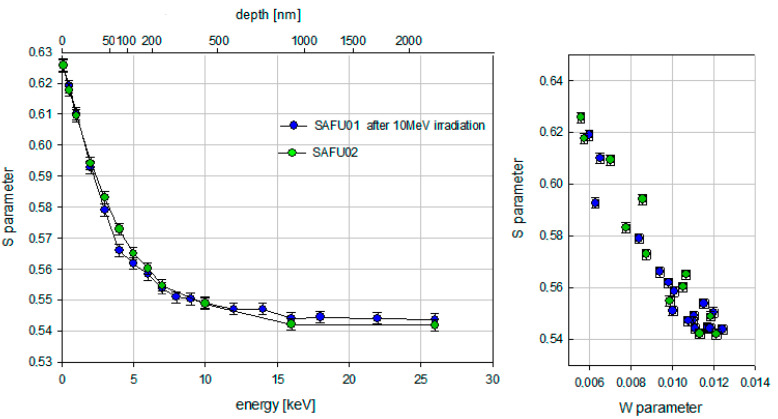
Results of Doppler spectroscopy after electron irradiation (10 MeV, 1 × 10^16^/cm^2^).

**Figure 8 materials-16-00203-f008:**
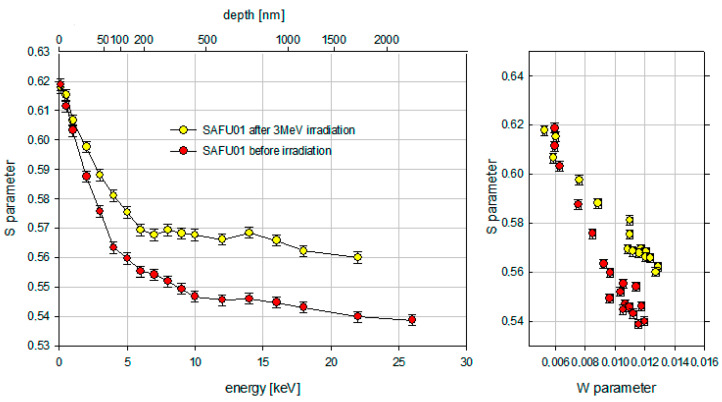
Results of Doppler spectroscopy after electron irradiation (3 MeV, 1 × 10^18^/cm^2^).

## Data Availability

Not applicable.

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
