# Peer review of "Investigation of Nitrogen and Vacancy Defects in Synthetic Diamond Plates by Positron Annihilation Spectroscopy"

_materials, 2022, doi:10.3390/ma16010203_

Round 1
Reviewer 1 Report
This work demonstrated the investigation of Nitrogen and Vacancy defects in diamond plates by PAS method. I believe this work provides a factile way to examine the quality of diamond-based materials. I spport the acceptance of this manuscript after minor revision.
1. PAS seems to be a useful tool to study the defects in diamond. Can the authors provide more methods to examine the quality of diamonds in the introduction? and compare the advantages of different pathways.
2. To boarden the applications of PAS, please provide other materials suitable for PAS investigations, as examples.
3. In figure5,please provide scale bars in the image.
Author Response
We thank the respected reviewer for the comments that we took into account on the text of the article, in more detail:
1. Yes, we did it in lines 60-75.
2. Yes, we did it in lines 87-89.
Reviewer 2 Report
The paper presents interesting results on nitrogen and vacancy defects in synthetic diamond plates by positron annihilation spectroscopy as well as infrared spectroscopy. The manuscript has been modified carefully prior to submission to MDPI Materials. I noticed one important quality-enhancing correction, should be necessary prior to final publishing, i.e.
Say somthing why SAFU02 contains more defects than SAFU01, this is clearly indicated from results in Fig. 7 and Fig. 8, you see at E=0 keV incident positron energy, S value for SAFU02 is larger than that of SAFU01. This agrees well with IR results. But the author should point out which type of defect should be ascribed to such positron results. Is it A or C+ or N+ defect? which one is negatively charged or attractive to positron?
Author Response
We thank the referee for such a valuable remark; indeed, before irradiation and annealing, C defects predominate in the SAFU01 sample; after irradiation and annealing, A defects increase. The appearance of NV centers, neutral NV0 and negative NV−, primarily NV− (negatively charged and attracts a positron), arising under proper irradiation and annealing, should significantly change the parameters of positron annihilation. The S-parameter increases, as can be seen from the data in Fig. 7 and fig. 8. So far, we have not found a quantitative interpretation in the works of other authors, but the trend is obvious.
Reviewer 3 Report
Reviewer report on the paper "Investigation of Nitrogen and Vacancy Defects in Synthetic Diamond Plates by Positron Annihilation spectroscopy" submitted to MDPI Materials by Marat Eseev et al.
In the paper, the methods of infrared spectroscopy and Doppler-broadening of positron annihilation radiation, using monoenergetic positron beam, were combined to investigate defects in synthetic diamond. The paper deserve publication in Materials provided that authors satisfactorily respond to the comments in the items A to E listed below, of them item D being the most crucial.
A. At the end of Sect. "1. Introduction", authors say that they "studied defects in nitrogen doped synthetic diamond plates ...". If the doping was intentional(?) then doping procedure should be described and resulting type of diamond structure (Ib ?) given in Section "2. Experiment". How the doses of electron irradiation are expressed in the text: electrons per total area 4×4 mm or per cm_2 ? The latter seems to be the more frequent convention.
B. The beam spot diameter (5 mm) slightly exceeded the side size of the sample (4 mm). Thus a portion of positrons did not hit the sample and annihilated in the surrounding materials. What measures were taken to avoid or suppress such a potential contribution to the DB annihilation photopeak?
C. At the row no. 169 and next ones, and again in Sect. "6. Conclusions", authors say that "... the nitrogen concentration in the sample SAFU01 decreased from 220 ppm to 200 ppm." As the difference amounts about 10 % of the value only, it would be recommendable to show here the experimental uncertainties appeared as significantly lower than 10 %.
D. The analysis of Doppler spectroscopy data suffers from too much simplicism. For example, authors' conclusion that "... significant vacancy formation concentrated at depths from 200 nm to 2400 nm" cannot be regarded as fully justified. Apparently, a more profound approach to this task, based on the VEPFIT analysis, is highly desired. It could quantitatively characterize the measured S vs. E dependencies in terms of positron diffusion lengths and layer thicknesses and, among others, allow for an interesting comparison of the present results with earlier ones, e.g. Fischer CG et al. (1999) or Uedono A. et al. (1999).
E. The reviewer is not a native English speaking person, but there is feeling that some language checking could be recommended.
Author Response
Dear reviewer, thank you for your comments.
Response A: Resulting type of diamond structure is Ib. Dose – 1018 electrons per cm2
Response B: The detectors were directed to the samples. Positrons that flew beside the sample annihilated outside the scanned area. minimally affecting result they annihilated outside the scanned area. They have invested minimal impact on the outcome.
Response D: In accordance with the comments – the VEPFIT analysis were added.
We added all changes to the article.
Round 2
Reviewer 3 Report
Authors satisfactorily responded almost all reviewer's comments. Only one additional comment may be suggested for optional considering to authors. The statement in Conclusions that "... significant vacancy formation concentrated at depths from 200 nm to 2400 nm", seems to be somewhat stronger than implied by the VEPFIT analysis based on the one-layer model used (i.e., is the number of defects below 200 nm lower than above 200 nm ?). Such a statement also slightly contradicts with the last statement added to preceding Section 5 in the revised text : " ... which stands for an increased number of defects below the surface" (i.e., the number of defects below 200 nm is not lower than above 200 nm ?). This happen, in particular, due to word "concentrated" which can easily be omitted.